# A Robust Automatic Ultrasound Spectral Envelope Estimation

**Jinkai Li**, **Yi Zhang, Xin Liu, Paul Liu, Hao Yin * and Dong C. Liu**

College of Computer Science, Sichuan University, No.24 South Section 1, Yihuan Road, Chengdu 610065, China; smilelijinkai@gmail.com (J.L.); zhangyiabby714@gmail.com (Y.Z.); tianzhilishu@gmail.com (X.L.); paulsliu@126.com (P.L.); dongcliu@163.com (D.C.L.)

*   Correspondence: yinhao@scu.edu.cn; Tel.: +86-135-5127-7539

**Abstract:** Accurate estimation of ultrasound Doppler spectrogram envelope is essential for clinical pathological diagnosis of various cardiovascular diseases. However, due to intrinsic spectral broadening in the power spectrum and speckle noise existing in ultrasound images, it is difficult to obtain the accurate maximum velocity. Each of the standard existing methods has their own limitations and does not work well in complicated recordings. This paper proposes a robust automatic spectral envelope estimation method that is more accurate in phantom recordings and various in-vivo recordings than the currently used methods. Comparisons were performed on phantom recordings of the carotid artery with varying noise and additional in-vivo recordings. The accuracy of the proposed method was on average 8% greater than the existing methods. The experimental results demonstrate the wide applicability under different blood conditions and the robustness of the proposed algorithm.

**Keywords:** ultrasound; power spectrum; velocity estimation

## 1. Introduction

Ultrasonography is widely used in clinical diagnosis due to its low cost, non-invasive nature, real-time imaging capability and the gradual increase of image quality. Nevertheless, there are also a number of shortcomings in ultrasound image, including ambient noise caused by environment, acquisition noise caused by ultrasound equipment, the existence of background tissue and other organs and the anatomical influences of body such as breathing motion, and body fat [1]. Noise existing in ultrasound images makes clinical diagnosis difficult. Maximum velocity and minimum velocity, respectively, correspond to the maximum velocity envelope and the minimum velocity envelope. Both maximum velocity and minimum velocity envelope are often used in a quantity of clinical diagnostic applications. In the diagnosis of carotid stenosis, internal carotid artery (ICA) peak systolic velocity (PSV), which can be observed in Doppler spectrogram, is an important medical parameter to grade the ICA stenosis [2]. Blood flow volume assessment, which is calculated by the maximum velocity of blood and the vessel area, can be used in a number of clinical applications [3] such as cerebral blood flow assessment [4], arteriovenous fistula inspection [5], anesthesia [6] and fetal outcome assessment [7]. With the increasing use of ultrasonography in clinical diagnosis, a more effective and automatic method needs to be proposed to reduce the work of doctor and to allow easier use by undertrained or untrained users.

In the process of ultrasonic imaging, the scattering of ultrasonic wave by human tissue, the interference of scattering echo and the motion of organs can result in noise of the ultrasonic image [8]. The noise of ultrasonic image makes accurate envelope estimation difficult and can result in difficulty in diagnosis. According to the Doppler equation, a single blood velocity should give rise to a single Doppler frequency. However, in fact, a single blood velocity may result in a range of

frequencies, which causes a phenomenon called intrinsic spectral broadening [9]. The existence of intrinsic spectral broadening also has bad impact on envelope estimation.

Ultrasonic spectrum envelope is a curve used to represent how the maximum or minimum values change over time. The traditional envelope is drawn by the experienced sonographers, which is inefficient and requires a lot of manual intervention. Quantities of methods have been proposed to estimate the peak velocity of one signal at each time which is the basis of envelope estimation. The early methods such as modified threshold crossing method and hybrid method are very sensitive to SNR [10–12]. Currently, methods using spectral envelope estimation contain geometric method (GM) [12], modified GM (MGM) [13], signal noise slope intersection (SNSI) method [14], and modified signal noise slope intersection (MSNSI) method [3]. These current methods work better than the early methods [12,14–16]. Most of the existing maximum velocity estimation methods are based on integrated power spectrum (IPS) at each time. These envelope estimation techniques are mainly focused on using image processing methods [17–19] so these methods can be reproduced easily. Each existing method has its limitation and cannot work if negative blood flow velocity exists. GM performs well when its IPS is a match with these typical characteristic shapes. MGM can get a good result when the noise of ultrasonic image is relatively low. Both GM and MGM are prone to overestimate the maximum velocity of blood flow. The predefined parameters are used in SNSI and MSNSI so that they cannot work well in the condition of high noise.

A robust automatic spectral envelope estimation based on ultrasound Doppler blood flow spectrograms is proposed in this paper. The quadratic iteration algorithm (QIA) based on the IPS is proposed to estimate the spectral envelope, which works well even when negative velocity exists. To evaluate the proposed envelope estimation method in this paper, the QIA was compared with two existing methods: MGM and MSNSI.

All of these above envelope estimation methods are compared against the true curve, which is calculated by the parameters of the phantom set-up or the ideal curve, as drawn by an experienced clinician. Comparison experiments were conducted on phantom recordings to verify the accuracy of the QIA, on carotid artery with different noise levels to verify the robustness of QIA and on different blood flow conditions to verify the wide applicability of QIA.

This paper is organized as follows. In Section 2, the detailed description of the quadratic iteration algorithm is given. In Section 3, the performance of the quadratic iteration algorithm and existing methods during tests on diverse phantom recordings and various in-vivo flow spectrograms is presented. In Section 4, the discussion of the experiments is presented. Finally, conclusion are given in Section 5.

## 2. Algorithm Description

### 2.1. Quadratic Iteration Algorithm

The existing spectral envelope estimation methods based on the integrated power spectrum perform well under many conditions, nevertheless each method has limitations. A quadratic iteration algorithm based on the integrated power spectrum is proposed in this paper. The quadratic iteration algorithm has the following steps:

### 2.1.1. Step 1

$S(n)$ is the input signal, as shown in Figure 1a, and $P(n)$ is the corresponding IPS, as shown in Figure 1b. The IPS is calculated as:

$$P(n) = \sum_{i=1}^{n} s(i), n = 1, 2, ..., N \tag{1}$$

where $N$ is the length of input signal.

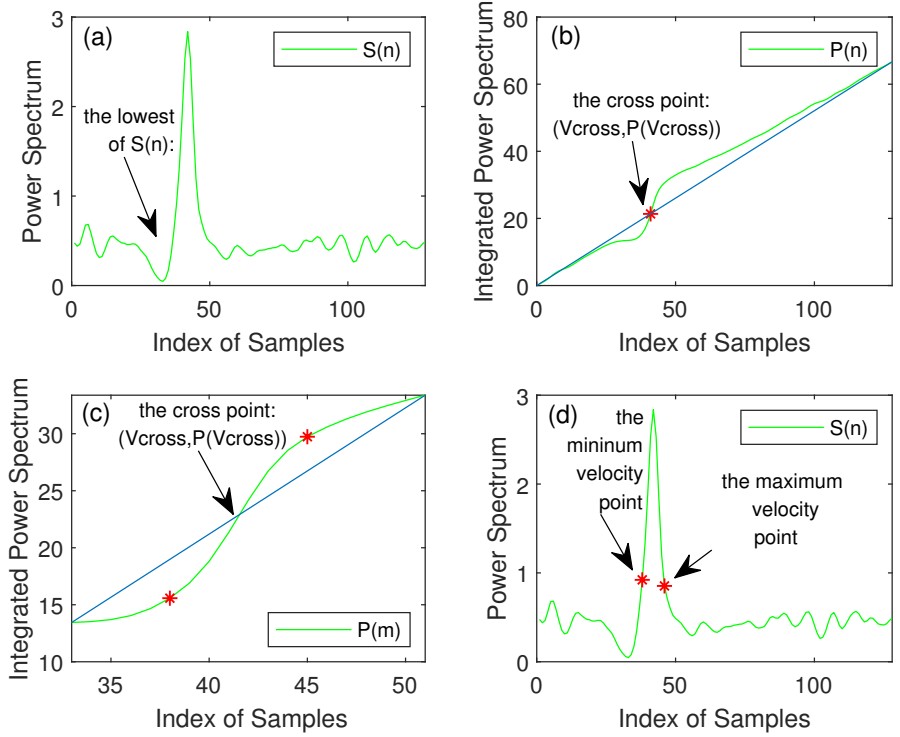

**Figure 1.** (**a**) The data of input signal; (**b**) integrated power spectrum (IPS); (**c**) new IPS after quadratic iteration algorithm; and (**d**) the maximum and minimum velocity points of the original signal.

#### 2.1.2. Step 2

Connect the start point and the end point of $P(n)$ to get a reference line, and the intersection points between $P(n)$ and the reference line can be obtained. The cross point $(V_{cross}, P(V_{cross}))$ is also the maximum energy point of the signal.

#### 2.1.3. Step 3

There are assumptions that the maximum flow velocity point $(V_{max}, S(V_{max}))$ and the minimum velocity point $(V_{min}, S(V_{min}))$ are in the vicinity of cross point. Thus, $P(n)$ accumulates a lot of meaningless noise, which can disturb the correct solution process. Equation (2) is used to cut the noise from $P(n)$ to get a new IPS $P(m)$:

$$P(m) = P(i), i = V_{cross} - \Delta V, ..., V_{cross} + \Delta V \tag{2}$$

$$\Delta V = V_{cross} - S_{lowest} \tag{3}$$

where $V_{cross}$ is the horizontal ordinate of the intersection point, $S_{lowest}$ is the horizontal ordinate of the minimum point that is searched from $S(1)$ to $S(V_{cross})$ and this point is shown in Figure 1a. Equation (2) means that both of the data from $S(1)$ to $S(V_{cross} - \Delta V)$ and the data from $S(V_{cross} + \Delta V)$ to $S(N)$ are considered as noise. Thus, $P(m)$ is a new integrated power spectrum, which is shown in Figure 1c.

#### 2.1.4. Step 4

Connect the start point of the $P(m)$ and the end point of $P(m)$ to get a new reference line, which is shown in Figure 1c. There are two important points in $P(m)$: the negative maximum distance from the connecting line and the positive maximum distance from the reference line. In practice, the positive maximum distance from the reference line is the maximum velocity point of this signal

and the negative maximum distance from the connecting line is the minimum velocity point of the signal. The position of this two important in $P(m)$ is shown in Figure 2.

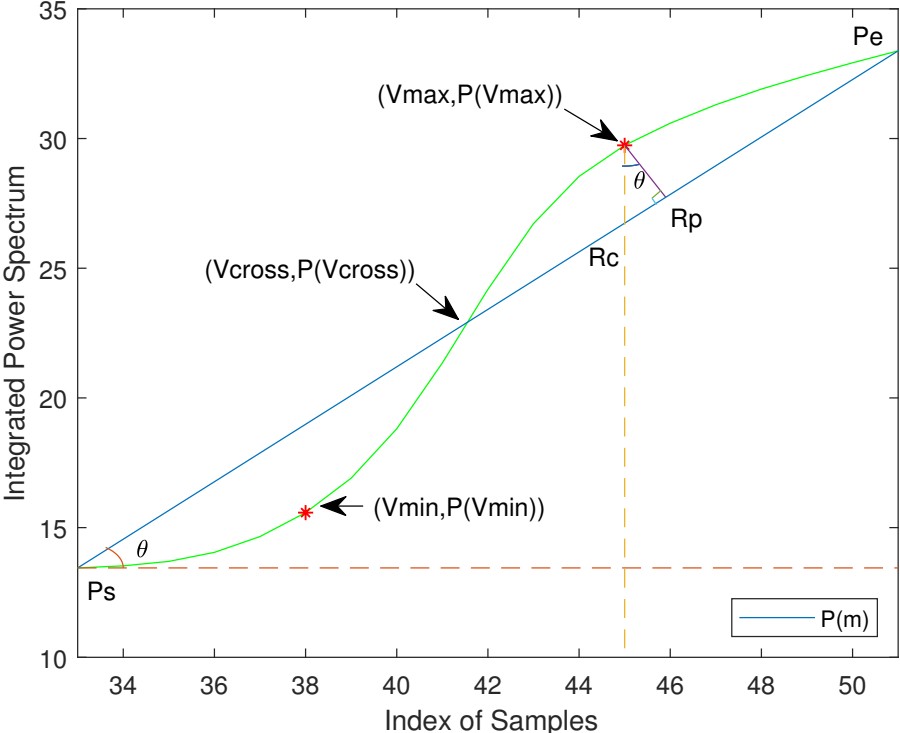

**Figure 2.** Location of the maximum velocity point and the minimum velocity point in P(m).

A suitable method to locate the maximum velocity point and the minimum velocity point is proposed in this paper. The entire algorithm is described as follows: As shown in Figure 2, the green line is the curve of $P(m)$, the start point of $P(m)$ is $Ps$, the end point of $P(m)$ is $Pe$ and the intersection of $P(m)$ with the reference line is $(V_{cross}, P(V_{cross}))$. Assuming $(x, P(x))$ is any point of curve $P(m)$. Drawing a line perpendicular to the x-axis through $(x, P(x))$ and the intersects with the reference line is the point $Rc$. $\theta$ is a constant value when $P(m)$ is determined by Equation (2). $a$ is defined as the distance between $(x, P(x))$ and $Rp$ and the distance between $(x, P(x))$ and $Rc$ is $c$. Thus, the equation $a = c * cos(\theta)$ is always established in Figure 2. $a$ is the biggest value when $c$ is biggest. Because $(V_{max}, P(V_{max}))$ is the farthest from the reference line, the distance between $(V_{max}, P(V_{max}))$ and $Rp$ is the maximum value compared with the other points of $P(m)$. The search of $(V_{max}, P(V_{max}))$ is transformed to the search of the maximum distance between the y-axis of $P(m)$ and the reference line.

Assuming the coordinate value of $Ps$ is $(Ps_x, P(Ps_x))$, the coordinate value of $Pe$ is $(Pe_x, P(Pe_x))$, and the coordinate value of any point of $P(m)$ is $(x, P(x))$. Thus, the distance of $P(i)$ and $Rc$ is calculated as Equation (4):

$$|P(x), Rc| = P(x) - \frac{(x - Ps_x)}{(Pe_x - Ps_x)}) * (P(Pe_x) - P(Ps_x)) - P(Ps_x) \tag{4}$$

The point $x$ where the value of $|P(x), Rc|$ is positive maximum value is the maximum value of the original point of $S(n)$ and the negative max value is the minimum value of the original point of $S(n)$. Both maximum velocity point and the minimum velocity point located in the corresponding original point are shown in Figure 1d.

All of the process described as above is called as the quadratic iteration algorithm (QIA). The QIA can not only avoid the calculation of trigonometric functions brought by coordinate rotation, but also reduce the computational complexity. The location of the maximum velocity point and the

minimum velocity point is easy to implement. The calculation of $V_{max}$ and $V_{min}$ only need to go through $P(n)$ twice, thus the computational complexity is only $O(2N)$ and it is very suitable for the real-time systems.

Connecting the $V_{max}$ point and the $V_{min}$ point together in the direction of the time axis, the spectrum envelope is acquired. Then, a smoothing algorithm is used to make the required envelope graphics clearer and reduce the burr-like artifacts.

### 2.2. Modified Geometric Method

This method is also dependent on IPS. Connect the start point and the end point of IPS to get the reference line. The maximum velocity point is estimated as the point in the IPS which is farthest from the reference line. MGM is the modified method of GM, however this method also has its deficiency. MGM is prone to overestimate the point of max velocity in the case of narrow-band signals. The method is shown in Figure 3.

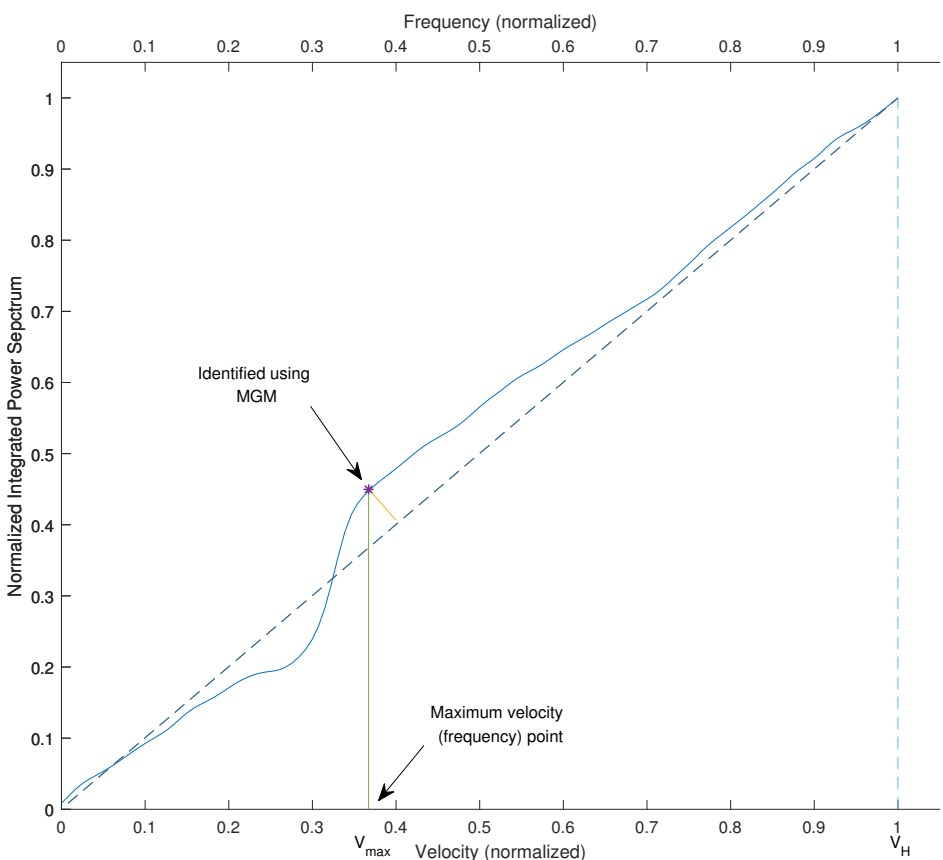

**Figure 3.** Location of the point of maximum velocity in the MGM.

### 2.3. Modified Signal Noise Slope Intersection

MSNSI is the modified method of SNSI. IPS is divided into three regions in this method. These three regions are the signal region, the knee region, and the noise region. All of these regions are shown in Figure 4. The signal region is defined as the full width at 70% of the signal peak in the power spectrum. $V_s$ is defined as the point of the end of the strong signal and $V_H$ is defined as the end point of the total signal. The knee center is identified based on an altered version of GM. It is located as the point on IPS that lies at a maximum distance from the reference line joining points on IPS corresponding to $V_s$ and $V_H$. The velocity of knee center is equidistant from the velocity corresponding to the end of the signal region and the start of the noise region. Thus, the noise start point can be

located when the knee center point and the end point of the signal region are defined. Two fit lines is used in this method: one line is in the signal region whose slope is $m_s$ and the other one is in the noise region whose slope is $m_n$. All of above points in IPS are described in Figure 4. The slope of the points in the knee region is calculated by the following equation:

$$m(x) = m_s x + m_n (1 - x) \tag{5}$$

where $x$ is the fractional signal contribution to the slope and $(1 - x)$ is the fractional noise contribution. In MSNSI, the maximum velocity point on IPS is heuristically found to the point where $x = 0.1$.

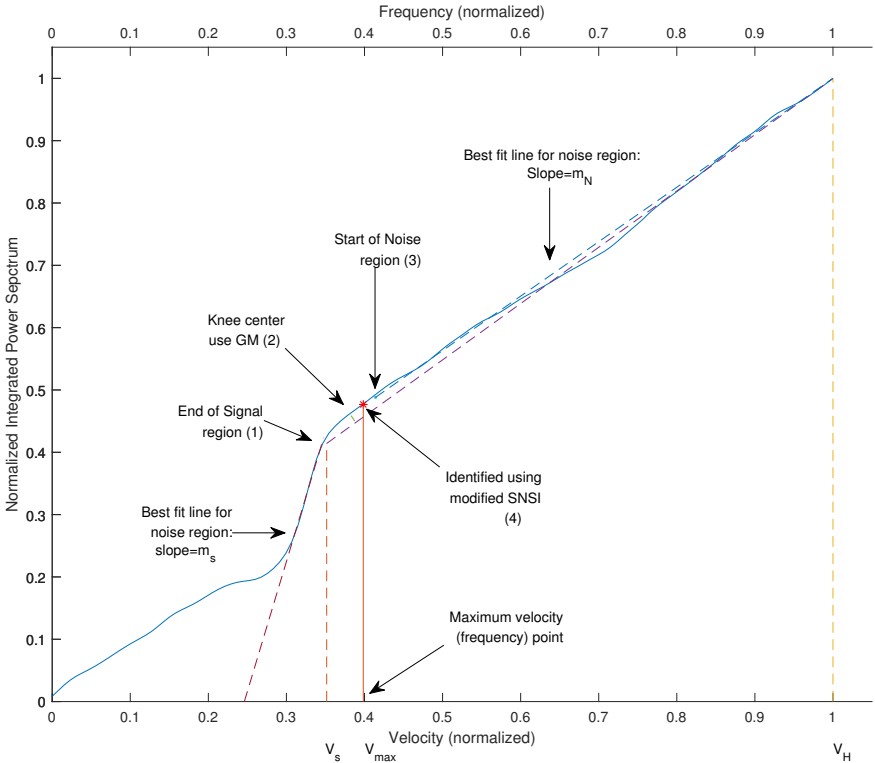

**Figure 4.** Location of the end of the knee center, signal region, noise start and noise end point in the MSNSI method. Steps (1)–(4) represent the procedure of MSNSI.

## 3. Experiments and Results

### 3.1. Data Acquisition and Processing

All of the methods in this paper are based on maximum velocity point detection. The maximum velocity point detection is dependent on the classic Doppler equation, which is shown as follows:

$$V_{max} = \frac{f_{max} * C}{2 * f_0 * cos(\theta)} \tag{6}$$

where $f_0$ is the center frequency of the transmitted ultrasound pulse, $C$ is the sound velocity and $\theta$ is the beam-to-flow angle. The maximum frequency point corresponds to the maximum velocity through this equation. Beam-formed in-phase quadrature (IQ) data of blood flow was derived from in-vivo data. Both datasets were collected from a Saset Insight 37C (Saset Healthcare Inc., San Francisco, CA, USA) machine. The entire sampling gate was split into 16 sub-gates. A hamming window function was applied to each sub-gate. Each of the window segments applied 128-point fast Fourier transform

(FFT) and the jump was 16. Each sub-gate obtained a single spectrogram, and then spectrogram compounding was used to get the entire final spectrogram.

### 3.2. Test Methods on Phantom Recordings

The in-vitro tests were carried out through the experimental set-up, KS205D-1 type Doppler phantom and imitation blood flow control system developed by Institute of acoustics, Chinese Academy of Sciences. This set-up can control the blood environment very well. With this set-up, the performance of the above algorithms can be verified. Figure 5a,b illustrates the true curve of envelope and the envelopes calculated by the MGM [13], MSNSI [3] and QIA methods using steady phantom flow. Figure 5c,d shows the envelope of true curve and results using above three methods for pulsatile phantom flow.

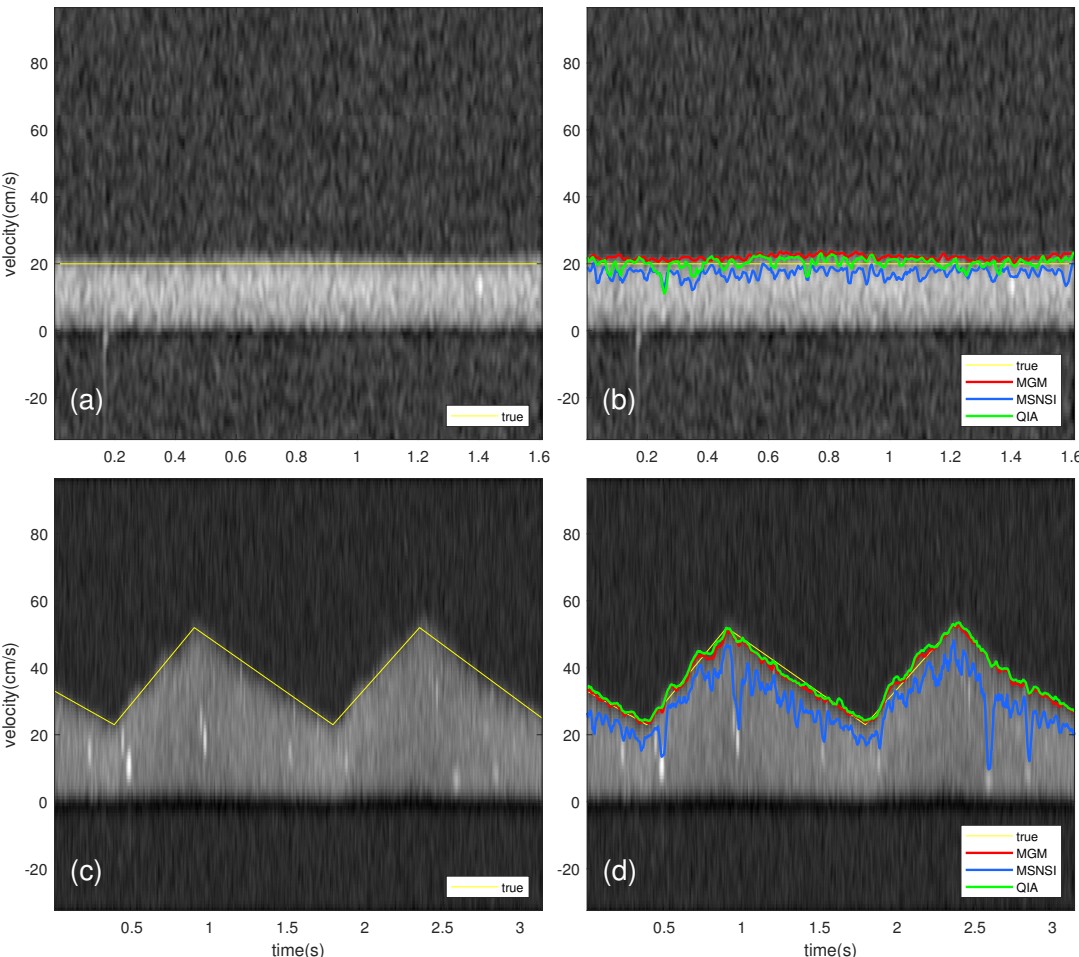

**Figure 5.** True curve and results using MGM and MSNSI, QIA methods tested on steady phantom flow and pulsatile phantom flow. (**a**) the spectrogram of steady phantom flow with true curve; (**b**) the results of methods tested on the spectrogram of steady phantom flow; (**c**) the spectrogram of pulsatile phantom flow with true curve; (**d**) the results of methods tested on the spectrogram of pulsatile phantom flow.

The true velocity of the steady phantom flow was set to 20 cm/s. The maximum velocity and the minimum velocity of pulsatile phantom flow were set to 62 cm/s and 33 cm/s, respectively. The error in $V_{max}$ estimation of results was bias and standard deviation were estimated as:

$$E_i = (V_{results})_i - (V_{true})_i \qquad (7)$$

$$bias = \frac{1}{N} * \sum_{i=1}^{i=N} E_i \tag{8}$$

$$\delta = \sqrt{\frac{1}{N-1} \sum_{i=1}^{i=N} \left| E_i - \frac{1}{N} * \sum_{i=1}^{i=N} E_i \right|^2} \tag{9}$$

where $(V_{results})_i$ is the maximum velocity estimated by above methods at any time point i and $(V_{true})_i$ is the true maximum velocity at that time point. If this phantom recordings is steady simulated flow, $(V_{true})_i$ is a constant. N is the number of time points. In the case of pulsatile flow, the bias and standard deviation are shown for the peak systolic points.

SNR is a value to evaluate the intensity of signal in the image. SNR is given by the following equation:

$$SNR = 10 * log \frac{P_s}{P_n} \tag{10}$$

where $P_s$ is the mean power contained in a region containing only signal and $P_n$ is the mean power contained in a region containing only noise. To test the robustness of envelope estimation in the spectrogram at different noise level, experiments were done by adding noise to the phantom blood flow. Spectral noise was added to the phantom flow spectra yielding signals with spectral SNR ranging from 10 dB to 4 dB. SNR of each case was calculated by Equation (10). Experiments on 10 sets of data were carried out to obtain the mean bias and standard deviation ($\delta$) of experiment values under different noise conditions. Table 1 shows the percentage values of bias of results compared with the true value by varying the added noise and Table 2 shows the mean standard deviation of the same data. Table 1 shows that the accuracy of the proposed method was increased by 8% compared on average with the existing methods in most instances.

**Table 1.** The mean bias of normalized results varying the added noise.

| Method Type | No Added Noise (SNR = 10 dB) | Noise Level 1 (SNR = 8 dB) | Noise Level 2 (SNR = 6 dB) | Noise Level 3 (SNR = 4 dB) |
|---|---|---|---|---|
| QIA | −0.34% | 1.45% | 4.46% | 5.2% |
| MGM | 9.83% | 9.96% | 11.05% | 14.97% |
| MSNSI | −14.03% | 13.79% | −14.19% | −15.03% |

**Table 2.** The mean standard deviation ($\delta$) of normalized results varying the added noise.

| Method Type | No Added Noise (SNR = 10 dB) | Noise Level 1 (SNR = 8 dB) | Noise Level 2 (SNR = 6 dB) | Noise Level 3 (SNR = 4 dB) |
|---|---|---|---|---|
| QIA | 3.06% | 3.41% | 3.15% | 6.42% |
| MGM | 3.61% | 3.64% | 4.10% | 7.79% |
| MSNSI | 7.39% | 7.12% | 6.48% | 8.61% |

### 3.3. Evaluate the Robustness of QIA

The envelope obtained from the proposed QIA, MGM and MSNSI were compared with the spectrum envelope drawn by an experienced clinician manually. Those above methods were tested on the spectrum of carotid artery and then noise was added to the ultrasound image to test the robustness of these methods. Using in-vivo data, Figure 6a illustrates the envelope obtained from the original signal without adding noise, while Figure 6b,d show the envelope estimation result of ultrasound image with the increasing noise added on the original signal respectively. SNR in Figure 6a–d is 21 dB, 19 dB, 18 dB and 17 dB, respectively.

As presented in [20,21], the figure of merit (*FOM*) is a value to evaluate the degree of similarity between two lines. In this study, the value of *FOM* was used to evaluate the similarity of the estimated envelope and the ideal curve. *FOM* is defined by the following equation:

$$FOM = \frac{1}{max(N, M)} \sum_{i=1}^{M} \frac{1}{1 + \alpha d(i)^2} \tag{11}$$

where $d(i)$ is the distance between predicted edge point $i$ and the nearest true edge point, $M$ is the actual number of estimated velocity points, $N$ is the number of standard velocity points and $\alpha$ is a positive weight factor. In general, $\alpha$ is a weighting factor for detection localization, which is a small constant and it is set to $1/9$ [21]. The value of FOM ranges from 0 to 1, and higher values indicate predictions with better quality detection. FOM was calculated on the envelope estimated by MGM, MSNSI, and QIA, and the results are shown in Figure 7. With the noise increasing on the ultrasound image, the FOM value of MGM declined severely. The FOM of QIA was still rather better than MGM and MSNSI even in the bad quantity of ultrasound image.

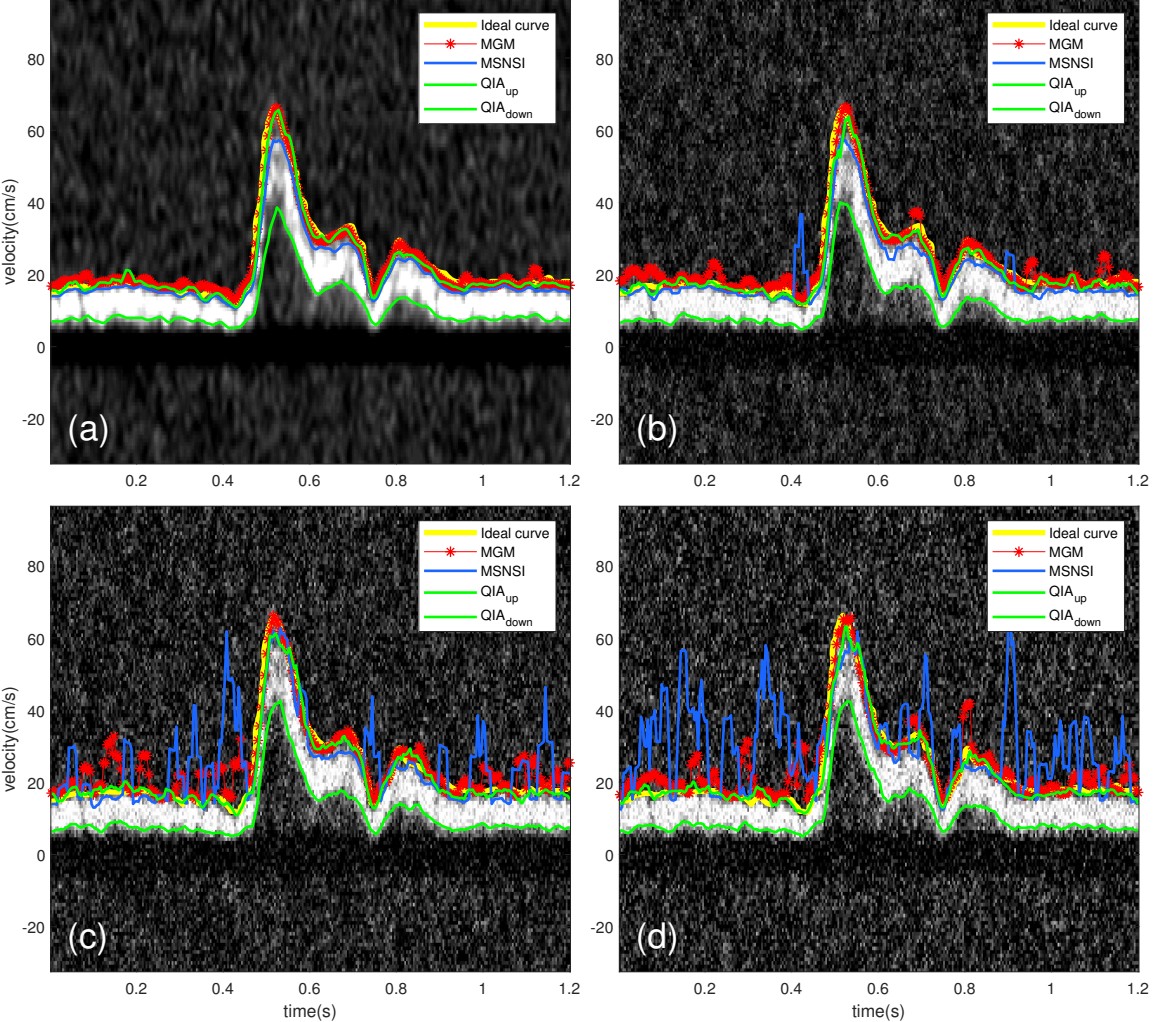

**Figure 6.** Comparison of the spectral estimation using MGM, MSNSI and QIA and ideal curve drawn by an experienced clinician, on a carotid artery spectrogram, in different noise conditions: (**a**) the results of methods tested on the original spectrum with no added noise; and (**b**–**d**) the results of envelope estimated when the level of noise is increasing.

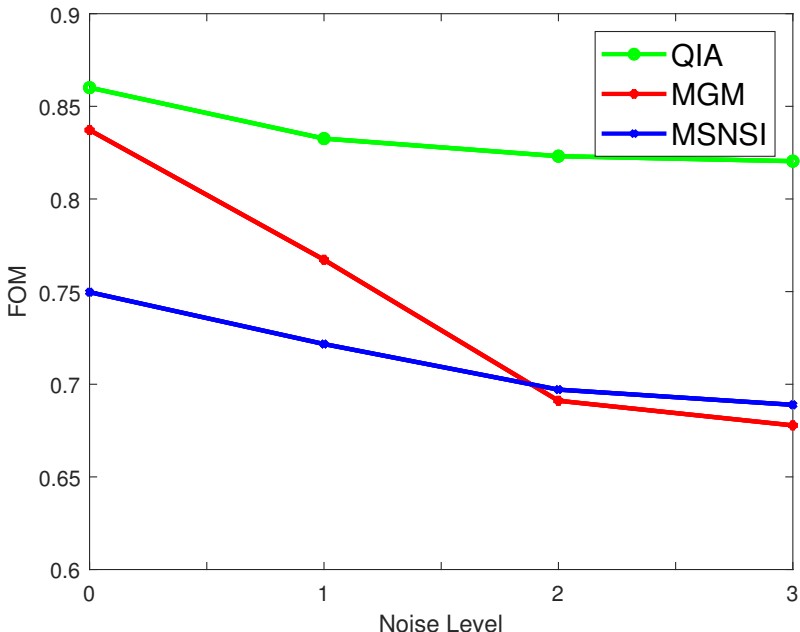

**Figure 7.** FOM of algorithms QIA, MGM, and MSNSI by varying variance of Gaussian white noise. No added noise is defined as Noise level 0, while Noise levels 1–3 represent increasing levels of Gaussian white noise.

*3.4. Test Methods on Different In-Vivo Recordings*

To evaluate the practicability of the proposed QIA, four comparison experiments were conducted on the spectrum of carotid artery, finger blood, kidney blood and heart blood. The comparison results are shown in Figure 8a–d, respectively. The QIA can locate the minimum velocity and the maximum velocity of the single signal. The minimum velocity point was used twice in finding the negative maximum velocity and the positive maximum velocity. In the case of existing negative blood flow, the QIA was applied twice. The first step was getting the minimum velocity point from the modified IPS(P(m)), which is the integrated value calculated from the start point of the input signal to the end point of input signal. The second step was getting the minimum velocity point from the modified IPS(P(m)), which is the integrated value calculated from the end point of the input signal to the start point of input signal. A FOM dataset is presented in Table 3 to show the difference between the calculated curve by the proposed QIA, MGM, and MSNSI and the ideal curve. As shown in Table 3, the FOM of the proposed QIA method is higher than other methods in all tested conditions.

**Table 3.** FOM of different methods tested on different blood setting.

| Method | Carotid Artery | Finger Blood | Kidney Blood | Heart Blood |
|--------|----------------|--------------|--------------|-------------|
| QIA    | 0.8601         | 0.6665       | 0.5851       | 0.6487      |
| MGM    | 0.8465         | 0.6646       | 0.0531       | 0.4180      |
| MSNSI  | 0.7497         | 0.3434       | 0.0258       | 0.1463      |

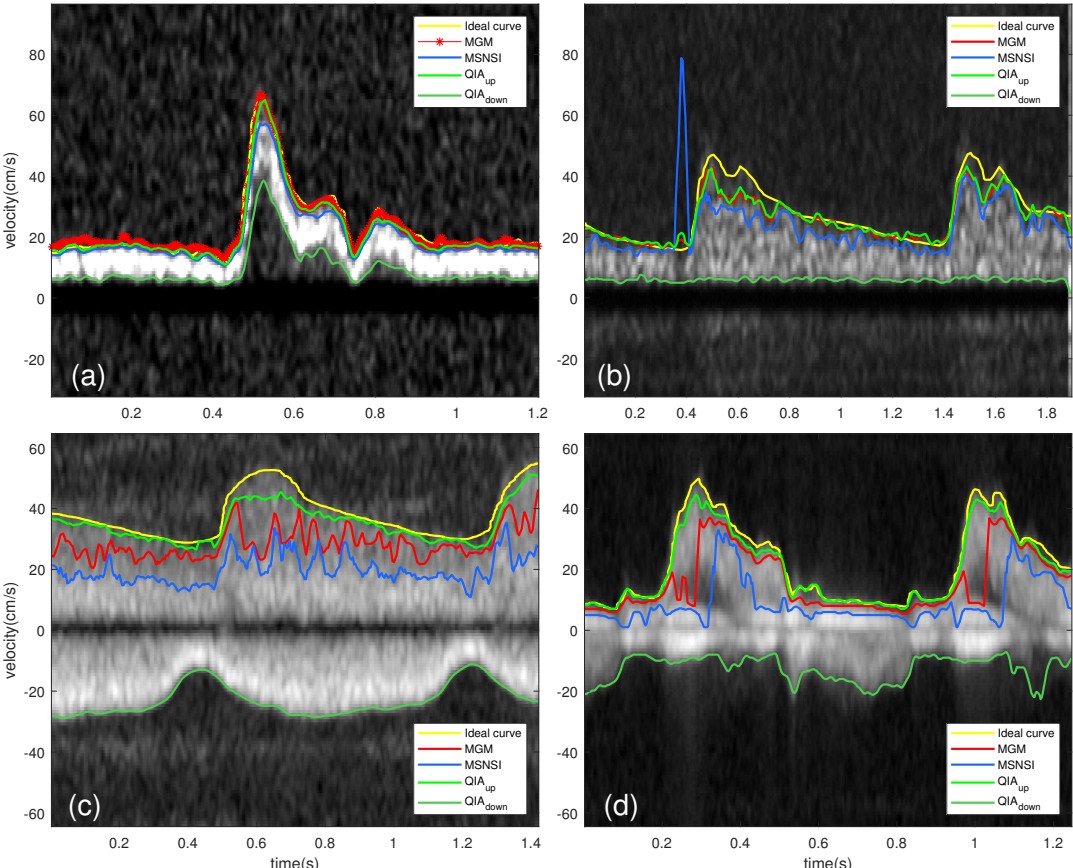

**Figure 8.** Envelope estimation results on: spectrum of carotid artery (**a**); finger blood (**b**); kidney blood (**c**); and heart blood (**d**).

## 4. Discussion

The existing methods MGM and MSNSI and the proposed QIA were tested on phantom flow with varying the added noise. As shown in Tables 1 and 2, MGM is easy to overestimate the point of maximum velocity in the case of narrow-band signals. Predefined parameters are used in MSNSI, thus this method is data dependent.

Using in-vivo data, Figure 6a–d shows that the QIA could still work well when high noise was added on the original ultrasound image. With the increase of noise, the performance of MGM and MSNSI worsened. Especially MSNSI almost did not work when the level of noise was too high, as shown in Figure 6d. Figure 8a–b expresses that the QIA works better in the carotid artery and finger blood with some noise. Figure 8c,d shows that, in the condition of kidney blood and heart blood, the MGM and MSNSI almost did not work for the existence of negative blood flow, while the proposed QIA could also work well. Even though the FOM of MGM in the condition of heart blood spectrum was 0.4180, which was not as bad as in the blood flow of the kidney, it is easy to see in Figure 8d that MGM did not work sometimes.

Quantitative analysis of the performance of the proposed QIA as well as MGM and MSNSI in different blood setting showed that the algorithm presented in this paper is accurate, practical and robust. The performance of QIA when compared with existing methods (MGM and MSNSI) showed that the QIA supports more anatomical applications than other methods. Both the above methods could work well in the original spectrum without added noise. The computational complexity of MGM and MSNSI was O(N) while the computational complexity of QIA was O(2N). The limitation of QIA was that it needed more time than MGM and MSNSI, even though the spectrum had a high SNR.

Future work on QIA methods includes improving the accuracy and developing a systematic method to realize the automatic measurement in the envelope.

## 5. Conclusions

This paper proposes a robust automatic spectral envelope estimation based on ultrasound Doppler blood flow spectrograms.The QIA methods was tested on phantom recordings to show the accuracy of the QIA. The QIA method was tested on an ultrasound image of carotid artery with several levels of noise. A strong robustness of the QIA was observed in the in-vivo recordings. The FOM of other method was low in the case of increasing noise, whereas the FOM of QIA was significantly higher in these cases. Experiments on different blood conditions showed that the QIA has a wide practical usage range even in the presence of negative velocity blood flow. The QIA method can be used for estimation of power spectrum envelope, and is practical and robust for a wide range of noise levels and different diagnostic applications. It is an important part of automatic measurement in ultrasound to approach the ultimate goal of intelligent ultrasound. As part of the growing trend towards intelligent ultrasound, this technique is expected to be beneficial for inexperienced ultrasound users and to decrease workload of all users.

**Author Contributions:** Conceptualization, J.L. and X.L.; Investigation, J.L. and X.L.; Methodology, J.L. and X.L.; Software, J.L.; Supervision, H.Y. and D.C.L.; Validation, J.L.; Writing—original draft, J.L.; and Writing—review and editing, Y.Z., P.L. and H.Y.

**Funding:** This work was supported by Department of Science and Technology of Sichuan Province (Grant No. 2016JY0084) and NSFC (Grant No. 81570848).

**Acknowledgments:** I would like to express my gratitude to all those who helped me during the writing of this manuscript. I also owe a special debt of gratitude to the related editor and reviewers for their valuable comments and suggestions to improve the quality of this paper.

**Conflicts of Interest:** The authors declare no conflict of interest.

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
