# Peer review of "A Robust Automatic Ultrasound Spectral Envelope Estimation"

_information, doi:10.3390/info10060199_

Reviewer 1 Report

There are still some spelling errors existing. Please take care of those. E.g. "Index Of Samples" should not have the "O" capitalized, words like "in-vivo", "in-vitro", "in-silico" etc. should be italicized, the word robust is misspelled as "robust0z" etc. are among a few.

Author Response

Comment No.1:
There are still some spelling errors existing. Please take care of those. E.g. "Index Of Samples" should not have the "O" capitalized, words like "in-vivo", "in-vitro", "in-silico" etc. should be italicized, the word robust is misspelled as "robust0z" etc. are among a few.

Response:

Thank you for your valuable advice. We have amended “Index Of Samples” to “Index of Samples”. Please check Page 3, Figure 1 and Page 4, Figure 2. We have amended “in-vivo” to “in-vivo”. Please check Page 1, line 6, Page 1, line 8, Page 2, line 65, Page 7, line 144, Page 9, line 181, Page 11, line 212, Page 12, line 234. We have amended “in-vitro” to “in-vitro”. Please check Page 7, line 151. We have amended “roubust0z” to “robust”. Please check Page 11, line 222.

Reviewer 2 Report

The authors have answered to all my previous remarks and I advice for an acceptance for this proposed paper.

Author Response

Comment No.1:
The authors have answered to all my previous remarks and I advice for an acceptance for this proposed paper.

Response:

Thank you very much for your advice. We appreciate for your warm work earnestly. Your comments are all valuable and very helpful for revising and improving our paper.

This manuscript is a resubmission of an earlier submission. The following is a list of the peer review reports and author responses from that submission.

Round  1

Reviewer 1 Report

Overall this manuscript has valuable information and the work done is good but there are quite a few things that can be rectified/ improved.

Major Issues:

In order to establish the proposed method the authors need to check the algorithm in a wide variety and a number of subjects, diseased and otherwise.

After the proposed method has been applied to a wide range of subjects, the result needs to be presented with a strong statistical analysis support.

There is no dedicated discussion section where the authors compared their results against existing literature.

The authors did not mention any limitations of the proposed method. They need to address that issue. A method is only as strong as its limitations.

The manuscript needs to be checked by a professional technical writer and also by someone proficient in written English.

Minor Issues:

LINE 183-185:     The sentence is very confusing and incomprehensible. Needs to be stated clearly.

LINE 16:                “the existing of” -> the existence of

LINE 31:                “result with” -> result in

LINE 55:                “spectral envelope which is also can work well if negative velocity exists” -> spectral envelope which works well even when negative velocity exists

LINE 58:                “against with the ideal” -> against the ideal

LINE 60:                “different blood conditions”, do the authors mean different blood flow conditions?

LINE 64:                “vivo” -> in-vivo

LINE 65:                Missing section number.

LINE 69:                “method has the” -> method has

LINE 70:                “limitation” -> limitations

LINE 148:              “vivo” -> in-vivo

LINE 153:              “almost not works” -> almost does not work

LINE 154:              “evaluate of the degree” -> evaluate the degree

LINE 157:              “α is small constant” -> α is a small constant

LINE 169:              “use case” -> step

LINE 169:              “which is” -> which is the

LINE 171:              “use case” -> step

LINE 171:              “which is” -> which is the

LINE 174:              “The case is also happened” -> The same thing happened

LINE 179:              “other methods not” -> other methods do not

LINE 186:              “in all the conditions” -> in all tested conditions

LINE 188:              “MGM is not work” -> MGM does not work

LINE 202:              “users and the decrease” -> users and to decrease

LINE 211:              “thesis” -> manuscript

Reviewer 2 Report

The article deals with Ultrasound spectral envelope estimation.

In the introduction, the authors describe the state of art and explain why their method can improve analysis of Doppler signals for undertrained or untrained praticians.

In the second section, the authors describes their method and two others methods (GM and SNSI) which will be used for comparison.

In the third section, the authors shows the results and compare them to the values deduced by a trained pratician and from other methods. They show a huge improvement compared to others methods.

Finally, the authors give a conclusion of their works.

After reading the paper, it seems that some improvements are needed for this paper even if the results are interesting.

Remarks on the paper:

1/ The authors say that they compared their work with other methods but some of them have be improved, as the SNSI in 2016 (10.1109/TUFFC.2016.2587381). Maybe they should take account of these new methods.

2/ The authors never give the relationship between frequency and velocity. Moreover they speak about velocities on the spectrum figures even if it is clearly frequencies. The figures have to be changed.

3/ In the equation 1, the star should be after the s and not after s(i). It can be noted that this notation is often used for the conjugate and can lead to misunderstanding.

4/ The usage of µ seems to increase the noise (it seems to normalise the noise as an average white noise) but the authors say in the paper that it decrease the noise impact. It should be explained by the authors or by adding a reference.

5/ the equation 4 in this form is quite obscure to me. It seems to be a windowing but I really think that the form of the equation has to be improved.

6/ the equation 6 mix two variables with the same meaning (x and i)

7/ I agree that the maximal value of Rc is equivalent to the maximal value of Rp but it should be quickly explained.

9/ the authors give us noisy spectrogram to prove that their method is robust but don’t give an idea of the SNR. I think that SNR value is important to prove the robustness of the method and help the reader to appreciate this method.

10/ the value of \alpha is set at 0.9 and not almost set at 0.9. Another thing is that the distance is often normalized and nothing is said about that in the paper.

11/ the authors should defined the noise level 0 as “no added noise” in the figure 6.

12/ the authors should give the velocities in figure 7.

Reviewer 3 Report

The manuscript, entitled “A robust Automatic Ultrasound Spectral Envelope Estimation” by Li et. al., proposed a new method to estimate the spectral envelope and compared with a couple of existing methods and showed an increase in accuracy.  I have the following comments for the authors to address.

Main problem I have on the study is the evaluation of the results.  The base value for comparison is the so called “idea curve”, which is supplied by an experience clinician. The process sounds very random, and clinician-dependent.  The results/conclusion could easily change if the idea curve was produced by a different clinician.  Hence, I don’t think the conclusion is decisive.  I will recommend the authors to test the method on a flow whose velocity distribution is given.  This experiment should be performed in vitro under very-well controlled environment.  Then the authors can further verify it with in vivo data. For human in vivo data, date from only one patient cannot be decisive, and experiment errors should be given based on multi-patient data.  

 Some minor mistakes I caught are listed below.  The English should be further improved for the future version, if the authors wish to revise and resubmit

QIA, IQ, needs to be defined at its first use.

Line 153:  Change to “It almost dose not work….”

Line 163: It should be “…in-vivo recording”